# Specific Immunoglobulin E and G to Common Food Antigens and Increased Serum Zonulin in IBS Patients: A Single-Center Bulgarian Study

**DOI:** 10.3390/antib11020023

**Published:** 2022-03-29

**Authors:** Milena Peruhova, Antoaneta Mihova, Iskra Altankova, Tsvetelina Velikova

**Affiliations:** 1Medical Faculty, Sofia University St. Kliment Ohridski, 1407 Sofia, Bulgaria; mmp@mail.bg; 2Laboratory of Clinical Immunology, University Hospital Lozenetz, Sofia University St. Kliment Ohridski, 1407 Sofia, Bulgaria; toni02@yahoo.com (A.M.); altankova@yahoo.com (I.A.)

**Keywords:** irritable bowel syndrome, IgE-mediated hypersensitivity, IgG-mediated hypersensitivity, zonulin, leaky gut

## Abstract

Irritable bowel syndrome (IBS) is a common functional gastrointestinal disorder whose pathogenesis is considered multifactorial, including abnormal gut motility, visceral hyperreactivity, psychological factors, disturbances in the brain-gut axis, leaky gut, oxidative stress, etc. We aimed to investigate serum levels of specific immunoglobulin E and G to common food antigens and zonulin and to assess their use in clinical practice for patients with IBS. Material and methods. We included 23 participants, 15 with IBS (diagnosed according to the Rome IV criteria) and 8 healthy controls. We investigated serum levels of specific IgG antibodies to 24 food antigens, specific IgE antibodies to 20 food antigens, anti-celiac antibodies, fecal calprotectin and serum zonulin by ELISA. Results. Food-specific positive IgG antibodies were significantly higher in patients with IBS than in controls (*p* = 0.007). IgE-mediated allergic reactions were found in five patients with IBS; no one had anti-TG antibodies. One-third of IBS patients demonstrated a low degree of chronic inflammation (positive fecal calprotectin test > 50 ng/mL) without specific bacterial infection. Serum levels of zonulin in IBS patients were higher than in healthy controls (0.378 ± 0.13 vs. 0.250 ± 0.14 ng/mL, *p* = 0.0315). However, no correlations between clinical symptoms and zonulin levels were found. Conclusion. The mechanisms of IgG hypersensitivity and low degree inflammation in IBS and elevated zonulin may contribute to multifactor pathogenesis in IBS.

## 1. Introduction

Irritable bowel syndrome (IBS) is a common functional gastrointestinal disorder in which pathogenesis is considered multifactorial [1]. Therefore, this condition has a wide range of symptoms, both digestive and extraintestinal. Moreover, IBS patients’ quality of life is significantly affected, and they may be unable to perform daily activities [2], which is why IBS is of great medical and socio-economic importance [3]. In line with this, due to the unclear causes of IBS, treatment is based mostly on lifestyle changes, diet, psychological sessions and treatment of concomitant symptoms by pharmacological means [4].

The suggested pathogenic mechanisms include abnormal gut motility, visceral hyperreactivity, psychological factors, disturbances in the brain-gut axis, leaky gut, oxidative stress, etc. [1]. Furthermore, 20% to 65% of patients with IBS complain of adverse food reactions with multiple and unclear mechanisms. The incidence of these reactions appears to be higher than in the general population [5]. In recent years, the role of food intolerance in these patients, caused by immune and non-immune mechanisms, has been increasingly discussed [6]. Both immune and non-immune mechanisms are being discussed for causing food hypersensitivity in IBS patients [6]. Food intolerance, together with food hypersensitivity/allergies and other unusual food intake reactions, are classified as adverse reactions to food antigens [7]. Food intolerance also includes non-immunologically mediated adverse reactions, such as direct effects of pharmacologically active food ingredients (e.g., tyramine, caffeine) and enzyme deficiencies (e.g., lactose and fructose intolerance). In contrast, food allergy (hypersensitivity) is used to describe the condition, mainly mediated by IgE antibodies that can be detected (e.g., allergy to cow’s milk protein, peanuts, soy) [8].

Dainese and colleagues [9] reported that more than 50% of patients with IBS were sensitive to food or respiratory allergens but could not identify which ones precisely and without typical clinical demonstrated and proven allergy. Increasing data show that IgE-mediated reactions are rare in this condition [9]. However, food adverse effects in patients with IBS may be due to non-IgE immune responses.

Recent studies suggest that such reactions may be mediated by IgG antibodies that occur later after exposure to a specific antigen. It has been found that IgG antibodies to food antigens may be normal, and their titers may vary with age. On the other hand, elevated levels of specific IgG antibodies have been observed in patients with asthma, hay fever, eczema and atopic dermatitis, and separately, IgG4 antibody levels are higher than average in patients with eczema and/or asthma [10]. However, IgG4 antibodies may be a physiological response of the intestinal immune system to antigenic food presentation [11,12]. The presence of IgG antibodies to specific food antigens is considered a protective mechanism of the organism and, therefore, should not be an indicator of the disease, i.e., not of diagnostic interest. In addition, data show that IgG antibodies against food antigens can be detected in healthy people [13]. Studies in healthy children have shown high levels of specific IgG antibodies to milk and eggs in the early years of life, whose titers gradually decrease with age [14]. In contrast, some studies demonstrated that IgG antibodies may play a role in mediating anaphylactic reactions [15,16].

However, when the intestinal mucosa is preserved and the oral tolerance is functioning, IgG antibodies against food antigens are minimal [17].

Interestingly, food-specific IgA antibodies could also be detected, usually in the gastrointestinal tract. They form complexes with the dietary antigens and are observed mainly in people with other diseases than food allergies, such as IgA-mediated nephropathy resulting in glomerulonephritis [18].

Moreover, more significant amounts of antigens and antibodies are required to affect this mechanism than in the case of IgE-mediated hypersensitivity. This suggests reconsidering the assumption that elevated titers of IgG/IgG4 antibodies to food antigens represent a “normal” response without clinical impact [19]. One of the most employed pathogenetic mechanisms for IBS is the “leaky gut” hypothesis [20]. Mixing together “hygiene hypothesis” for environmental and lifestyle changes to be too “clean”, as well as genetic background, increased gut permeability and changed microbiome, Fasano et al. described a “hyper-belligerent” immune system responsible for some chronic inflammatory diseases, including IBS [20]. However, gut permeability resulting from an initial cause for many autoimmune, infectious, metabolic and tumor diseases is still debated. In line with this, proteins from the zonulin family have been employed as critical players in modulating intestinal permeability.

Zonulin and intestinal fatty acid-binding protein (I-FABP) have been investigated as serum indicators for intestinal permeability. It is known that zonulin is a human equivalent of the toxin Vibrio cholerae zonula occludens, which is an endogenous regulator of the intercellular endothelial and epithelial tight junctions [21]. Zonulin has also been considered a modulator of both blood-brain and gut barriers based on its regulatory properties [22]. Thus, it is not surprising that it has been found elevated in celiac disease and even in type 1 diabetes [21].

Furthermore, the role of zonulin in gut permeability in celiac disease was confirmed by using larazotide acetate. This synthetic peptide inhibits zonulin and has been proven to prevent intestinal tight junction opening. In addition, a clinical study demonstrated that this inhibitor ameliorates symptoms in non-responsive celiac disease and confirms the relevance of zonulin in intestinal permeability once again [23].

The role of zonulin and I-FABP in IBS has not been well studied. It is assumed that when activated, the zonulin pathway is involved in mucosal homeostasis maintaining, but not always in pathological conditions [21]. Studies in mice models demonstrated that increased gut permeability indicated by increased zonulin levels is linked to only low-grade inflammation and usually normal architecture of the epithelial layers [24]. Therefore, zonulin pathways, together with genetic, immune, and microbiome functions and environmental triggers, may contribute to the pathological state of a “leaky gut” connected to many diseases.

On the background for possible immune mechanisms connected with IBS development, we aimed to investigate serum levels of specific immunoglobulin E and G to common food antigens and zonulin and assess their use in clinical practice for patients with IBS.

## 2. Materials and Methods

### 2.1. Subjects

In this single-center study, we included twenty-three participants. Fifteen patients with IBS were enrolled (3 males and 12 females at mean age, 36 ± 10 years) and 8 healthy controls. IBS patients were diagnosed according to the Rome IV criteria after excluding any other organic bowel disease in the Clinic of Gastroenterology of University Hospital “Lozenetz”. Rome IV criteria: recurrent abdominal pain on average at least 1 day per week during the previous 3 months that is associated with 2 or more of the following:-Symptoms related to defecation (may be increased or unchanged);-Symptoms associated with a change in stool frequency;-Symptoms related to a change in stool form or appearance.

All patients filled out a validated questionnaire about their symptoms and complaints (added to the anamnesis data). The criteria for excluding individuals (patients and healthy) from the study were: any clinically significant systemic disease, inflammatory bowel disease, immune deficiency, previous abdominal surgery or current pregnancy.

In addition, all participants (IBS patients and healthy controls) were invited to complete a questionnaire with data on eating habits, complaints from various organs and systems (if available), as well as to indicate whether they associated specific side effects (mild, moderate, severe) with particular foods.

All participants in the study were informed in detail about the purpose and methods used in the study and then signed their informed consent. The design and protocol of the study were approved by the Ethical committee of Lozenetz Hospital and Sofia University (No 80-10-160/25.04.2018).

### 2.2. Methods

Specific IgG antibodies to 24 food antigens were tested in serum samples from all subjects using an enzyme-linked immunosorbent assay with a commercial IgG Screen Nutritional 24 ELISA panel (ILE-SCG25; Immunolab GmbH, Kassel, Germany) according to the manufacturer’s instructions. The results obtained were presented as U/mL and reaction classes (<0.35 U/mL—class 0 (negative); 0.36–0.70 U/mL—class 1; 0.71–3.50 U/mL—class 2; 3.51–17.50 U/mL—class 3; 17.51–50 U/mL—class 4; 51–100 U/mL—class 5; >100 U/mL—class 6). Moderate positive results were considered class 1 + class 2 (with mild clinical manifestations and no required changes in the diet), and strongly positive—class 3 + 4 + 5 + 6 (where the symptoms are presented, and there is an improvement after dietary exclusion of a certain food. Our study used the units of reaction classes as reflecting a relative amount of specific IgG to a given food antigen. Twenty-four food antigens were egg white, cow’s milk, cod, wheat, rye, barley, rice, orange, banana, kiwi, strawberry, celery, soybean, carrot, tomato, garlic, hazelnut, peanut, curry, pepper, sesame, baker’s yeast, pork, beef. We chose this extended panel based on the previous studies for the most common food antigens investigated in IBS patients.

Specific IgE antibodies against food allergens were measured by a commercially available ELISA kit (EUROLINE Food IgE, Euroimmun, Lübeck, Germany), and included the following 20 food allergens: egg white, egg yolk, cow’s milk, baker’s yeast, wheat, rye, rice, apple, kiwi, apricot, celery, soybean, carrot, tomato, potato, hazelnut, peanut, almond, codfish, crab. The results were classified as follows: <0.35 U/mL—negative, class 0; 0.36–0.70 U/mL—class 1; 0.71–3.50 U/mL—class 2; 3.51–17.50 U/mL—class 3; 17.51–50 U/mL—class 4; 51–100 U/mL—class 5; and >100 U/mL—class 6.

Serum zonulin was assessed by a commercially available ELISA kit (Human Zonulin ELISA Kit, Cusabio, Wuhan, China). The detection range of the kit was 0.625–40 ng/mL, with a sensitivity of less than 0.156 ng/mL, intra-assay precision (CV%) <8 %, inter-assay precision (CV%) <10%, and a specificity—highly specific, however, with a declared cross-reactivity between human zonulin and all the analogs. However, some comments on the cross-reactivity and specificity of the used kit are discussed in the paper’s discussion section. We followed the manufacturer’s instructions. The final calculations were made after measuring OD values with a SPARK photospectrometer and generating a four-parameter logistic (4-PL) curve against the standards OD values.

Additionally, we performed testing for anti-tTG IgG and anti-gliadin antibodies with immunoblot (GAF-3X, EUROLINE Celiac Disease Profile IgG, Euroimmun, Lübeck, Germany), where the visual evaluation, EUROLineScan (Euroimmun, Lübeck, Germany) intensity and results were the following: No signal (0–5)—0, negative; very weak band (6–10)—(+) borderline; medium to a strong band (11–25) or (26–50)—(+ or ++) positive; and very strong band (>50)—(+++) strong positive result.

Fecal samples from study participants were examined by immunochromatographic methods for calprotectin (Calprotectin CerTest Biotec, Zaragoza, Spain); for Clostridium difficile toxin A + B (CerTest Biotec, Zaragoza, Spain); and for Helicobacter pylori (Pylori-K-SeT, Coris BioConcept, Gembloux, Belgium).

#### Statistical Analysis

The raw data were analyzed with the statistical analysis software packages SPSS, v.19 (IBM, Armonk, NY, USA) and Graph Pad Prism v.9. The percentage/frequency and degree of positive outcomes in IBS patients and healthy individuals were compared. Where appropriate, nonparametric t-tests were used to compare specific IgG levels in sick and healthy individuals. The differences were considered statistically significant at *p* < 0.05.

## 3. Results

IBS patients’ characteristics were as follows: nausea, vomiting, diarrhea, constipation, abdominal pain, flatulence, weight problems (100%); skin symptoms (urticaria, rashes, pruritus) (20% of the patients); different degrees of neurological complaints (migraine, headache, insomnia, impaired concentration, anxiety, fatigue) (73%); mucorrhea (27%). One-third reported different allergies (to drugs, pollens, wasp stings, two patients were with polyallergy) most probably related to the IgE pathogenetic mechanism. No recent significant intestinal infections have been reported in all participants. One-third of the IBS patients took different antibiotics during the last 3 months; 48.7% took probiotics during the previous 3 months.

We found that our IBS patients demonstrated significant levels of specific IgG to different food antigens (>3.51 U/mL): egg white (40%), 53,3% to cow milk (53.3%), wheat (33.3%), orange (20%), kiwi (33.3%), tomato (20%), garlic (26.7%), and hazelnut (26.8%). Overall, food-specific positive IgG antibodies levels (>0.35 U/mL) were significantly higher in patients with IBS than in controls (*p* = 0.007). More detailed results regarding IgG testing of this cohort of patients were reported previously [25,26].

In Figure 1, we present the distribution of specific IgG antibodies to food antigens among patients with IBS and healthy controls. Although two clusters are formed in the IBS patients (with higher and lower specific IgG than the median), no other associations or significance were found.

IgE-mediated allergic reactions were found in three patients with IBS (class I positive reaction to potato), one IBS patient (class I to peanut) and one patient (class II to crab). No one of the healthy people exerted positive IgE to food antigens. Although all IBS patients had various gut complaints, mucorrhea and skin symptoms, no one had anti-TG antibodies in the serum. However, one patient was positive for anti-GAF-3X antibody (weak result).

In 1/3 of the patients with IBS, we established evidence of low degree chronic inflammation (positive fecal calprotectin test > 50 ng/mL) without specific bacterial infection. We did not detect calprotectin in the healthy group.

When we compared the serum levels of zonulin between IBS patients and healthy controls, we found significantly increased levels in the patients (0.378 ± 0.13 vs. 0.250 ± 0.14 ng/mL, *p* = 0.0315) (Figure 2). However, no correlations between clinical symptoms and zonulin levels were found.

In Figure 3, we presented the distribution of zonulin levels compared to IgE and IgG positive results for common food antigens. However, we did not find significant correlations between zonulin levels and specific IgG antibodies (Figure 4).

## 4. Discussion

Our study design speculates that inflammation could be connected with the immune-mediated reaction of specific IgG and/or IgE to particular food antigens and zonulin pathway activation in the gut. Other mechanisms inducing temporary or permanent gut disturbances can be found in IBS patients. Indeed, we found significantly higher and more frequent food-specific IgG in the Bulgarian cohort of patients with IBS than healthy controls. Additionally, we observed two clusters in the IBS patients group—with higher and lower than the median IgG to specific food antigens, but without any other associations or significance.

However, we did not find evidence of IgE allergic or celiac autoimmune mechanisms playing a significant role in our IBS patients. The exact mechanisms of IgG hypersensitivity in IBS disease are not clear. It is possible that low degree inflammation in IBS is multifactor and depends on many other immune mechanisms in the gut.

Collecting a wide range of anamnestic data on patients’ subjective complaints possibly related to different foods, as well as various clinical complaints, allowed us to search for a link between anamnestic data and objective immunological results of specific IgG and IgE antibodies against different foods. Differences in the frequency of specific IgG antibodies between patients and healthy people unequivocally show higher IgG sensitization in patients with IBS against multiple antigens, although to varying degrees. In general, this was confirmed statistically, but due to the insufficient number of subjects, this association was not established for each food allergen.

At this stage, defining a pathogenetic mechanism is challenging. Similar to other authors, we did not find a correlation between subjective, anamnestic symptoms and laboratory testing of IgG hypersensitivity in the present study. However, this does not preclude the possibility of IgG antibodies involved in a possible pathogenetic mechanism. In this regard, we studied some markers of chronic inflammation in the feces in patients, such as fecal calprotectin and infections typical of other chronic inflammatory bowel diseases—Crohn’s syndrome and ulcerative colitis. In about 1/3 of the patients, we found calprotectin >50 ng/mL, indicating low-grade inflammation. Hypothetically, such inflammation may be associated with an immune-mediated response involving specific IgG and the formation of immune complexes with the respective food antigens.

Several epidemiological and uncontrolled studies have confirmed IgG hyperresponsiveness to dietary antigens in patients with IBS [27,28,29,30,31,32,33,34]. In addition, the role of specific IgG4 subclasses in food antigens was confirmed in childhood [27], adults with IBS [28,30,31] and those with migraine combined with IBS [34]. Some studies even found an association between food hypersensitivity and inflammation in the gut, obesity or functional disorders [33]. However, most studies demonstrated controversial and unconvincing results regarding eliminating diets in IBS and other conditions [29,32,34].

It is plausible that there is a participation of other immune mechanisms in the so-called food intolerance, given that changes in intestinal mucosa permeability by various stimuli, the antigenic load and antigen presentation in the mucosal immune system can theoretically increase [8]. For example, the presentation of antigens by dendritic cells leads to the activation of B lymphocytes producing IgE/IgG antibodies and T lymphocytes, which in turn secrete cytokines. In addition, the activation of mast cells and eosinophils leads to the release of various mediators. These events can lead to changes in the gut that are associated with low-grade inflammation and abnormal mucosal sensitivity in patients with IBS. In addition, IgG-food antigen complexes can spread to other tissues and organs and cause inflammatory reactions.

Eliminating foods to which patients have elevated IgG titers improves their symptoms. However, further research is needed to determine precisely in which patients’ IgG-food intolerance tests would play an essential role in deciding their diet [35,36].

Kvehaugen and co-workers failed to find a link between patients’ suspected milk and wheat intolerance and the corresponding antibodies (IgG and IgA) [32]. Therefore, the authors of this study recommend not using the results of IgG antibody testing for dietary recommendations in patients with morbid obesity and gastrointestinal complaints [32]. We also did not establish associations between the anamnestic data and the results obtained for specific IgG antibodies to the same foods.

Therefore, it is clear that there is no single answer to the question of whether a diet based on IgG tests has a positive role in patients with IBS. Critics of these methods, including the American Academy of Allergy, Asthma and Immunology (AAAAI), the Canadian Society of Allergy and Clinical Immunology (CSACI) and the European Academy of Allergy and Clinical Immunology (EAACI), claim that IgG antibodies against certain food antigens are a normal protective, physiological response of the body, and do not recommend the use of IgG tests to diagnose food allergies or food intolerance/sensitivity [19]. More studies are needed to determine the biological and clinical significance of elevated IgG antibody titers and whether they can be used as a basis for diet preparation.

Additionally, plenty of studies determined elevated zonulin levels in IBS patients. For example, Singh et al. measured zonulin and I-FABP in IBS patients with diarrhea-predominant and constipation-predominant forms compared with celiac disease patients and healthy controls [37]. They found significantly elevated zonulin in IBS compared to other groups. However, similar to our results, these increased levels did not correlate with the clinical picture.

Furthermore, Singh et al. discovered a strong connection between zonulin levels and IBS diarrheal form and bowel habit severity [37]. As a result, serum or stool zonulin levels, another non-invasive method for investigating the integrity of the intestinal barrier, might be regarded as a possible IBS molecular marker. Serum zonulin, a biomarker of defective intestinal permeability, was shown to be higher in individuals with diarrhea-predominant irritable bowel syndrome and constipation-predominant irritable bowel syndrome than in healthy controls, with levels equivalent to celiac disease. Furthermore, zonulin might be a simple serological biomarker for the changed intestinal permeability in IBS patients. Despite having normal small mucosal histology, IBS-D patients demonstrate tight junction dysfunction (as determined by zonulin levels) equivalent to active celiac disease. However, more research is needed to corroborate our findings of the role of zonulin in IBS. Identifying IBS patients with zonulin-mediated intestinal tight junction dysfunction may allow mechanistically focused IBS therapy (e.g., larazotide acetate) [37].

Furthermore, Caviglia et al. [38] explored the involvement of zonulin, a haptoglobin-2 precursor, in gut permeability and inflammation, taking into account its role in tight junction activity. Antigens might be easily delivered past the intestinal barrier, resulting in immunological activation [38].

Barbano et al. found no substantially increased levels of zonulin in IBS-D patients (*n* = 15) compared to controls in small research [39]. Some investigators, such as Prospero et al., also demonstrated a significant decrease in zonulin levels after a low-FODMAP diet [40].

Still, the precise causes of zonulin release in IBS are unknown. Enteric infections and gluten have been identified as the two most effective inducers of zonulin release [21]. Up to date, we know that zonulin was discovered to be pre-haptoglobin-2 (pre-HP2). Because of a polymorphism in the haptoglobin (HP) gene, represented by the HP1 and HP2 alleles, three distinct genotypes/phenotypes can be expressed: HP1-1, HP2-1, and HP2-2 [41,42]. Individuals with the HP1.1 genotype synthesis haptoglobin 1, whereas those with the HP2-2 genotype exclusively synthesize HP2 [43].

We have to emphasize other aspects of zonulin determination. Commercial zonulin enzyme-linked immunoassay (ELISA) kits have spared many costs to researchers, as well as time. Recent research, however, [44,45] has shown that these kits (for example, Immundiagnostik and Cusabio) do not detect zonulin but rather a haptoglobin and complement factor C3 [45]. Furthermore, other researchers have indicated that zonulin fails to detect diseases with known modest intestinal permeability [46]. In line with this, we want to state a few limitations of our study. Firstly, we did not have data on extraintestinal symptoms that might contribute to IBS. Secondly, we did not assess intestinal permeability with other methods such as immunohistochemistry for tight junction proteins. We did not also evaluate patients’ and controls’ HLA genotype or microbiome composition. However, we investigated gluten sensitivity by assessing two antibodies (anti-tTG and anti-GAF-X3). We also did not evaluate the IBS severity or bowel dysfunction. Last but not least, since a recent study demonstrated that some commercially available ELISA kits for zonulin (such as this used by us) could detect other proteins in the zonulin family, which are structurally and functionally related to zonulin, we discussed our results with caution and agreed that they need to be confirmed by other studies [44].

## 5. Conclusions

Our findings of significantly higher and more frequent food-specific IgG in the Bulgarian cohort of patients with IBS than healthy controls, along with no evidence of IgE allergic or celiac autoimmune mechanisms in our IBS patients, suggest that the mechanisms of IgG hypersensitivity may have a role in IBS pathogenesis. It is possible that low degree inflammation in IBS is multifactor and depends on many other immune mechanisms in the gut. Additionally, our results for increased zonulin levels confirm the investigations of other study groups. However, the role of zonulin in IBS pathogenesis has to be further evaluated.

## Figures and Tables

**Figure 1 antibodies-11-00023-f001:**
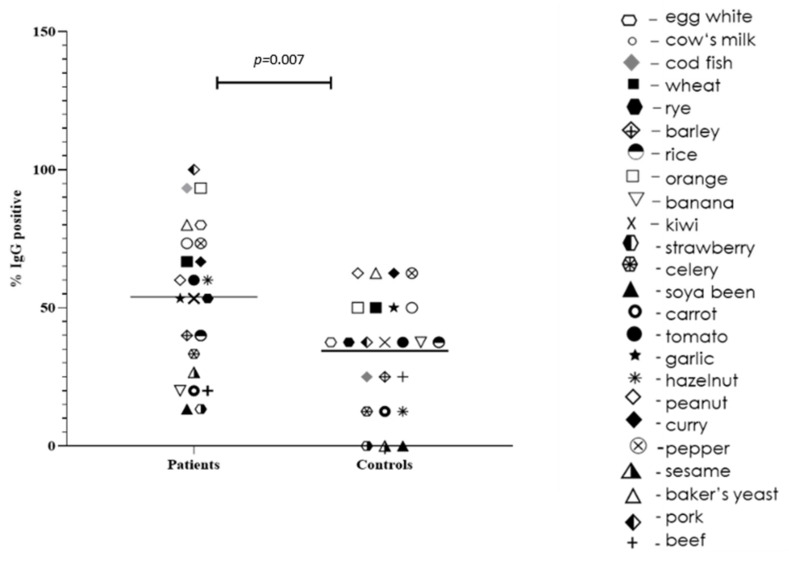
Positive IgG to food antigens between patients with IBS and healthy controls. Each antigen is represented by a separate symbol; the median for each group is represented.

**Figure 2 antibodies-11-00023-f002:**
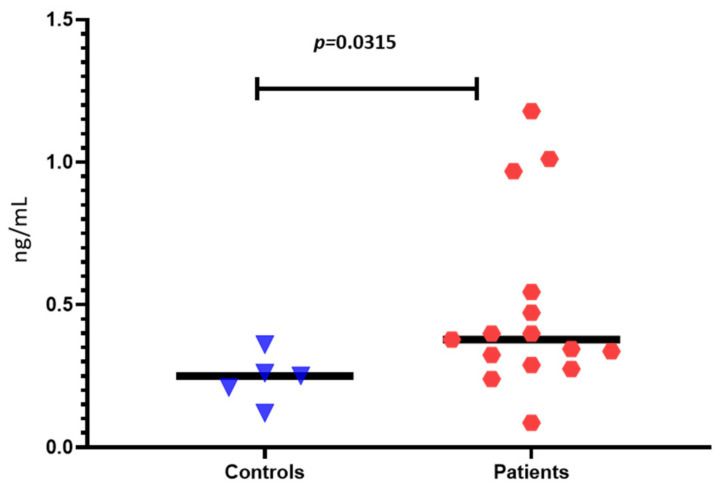
Zonulin levels of IBS patients and healthy controls.

**Figure 3 antibodies-11-00023-f003:**
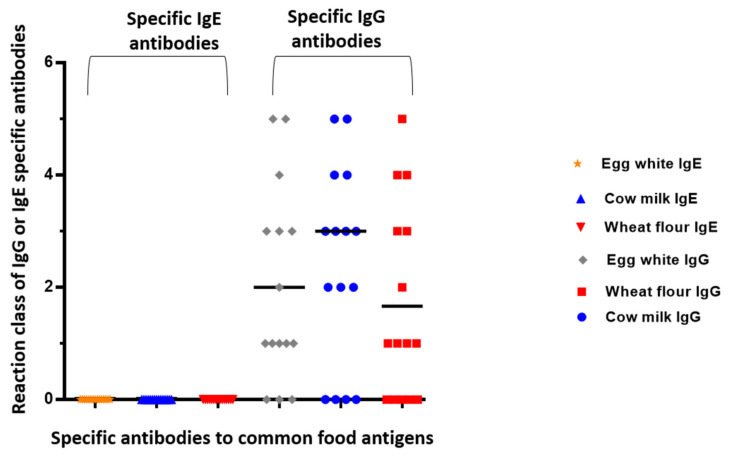
Distribution of positive IgG and IgE reactions against common food antigens.

**Figure 4 antibodies-11-00023-f004:**
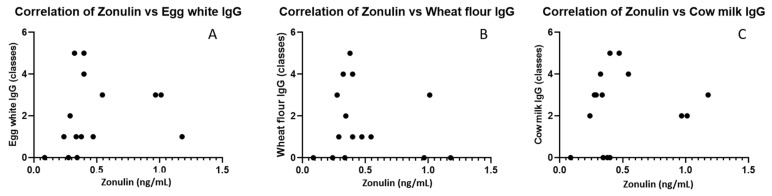
Correlations of zonulin levels with IgG to egg white (**A**), IgG to wheat (**B**) and cow’s milk (**C**).

## Data Availability

The datasets generated for this study are available on request to the corresponding author.

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
