# Peer review of "Specific Immunoglobulin E and G to Common Food Antigens and Increased Serum Zonulin in IBS Patients: A Single-Center Bulgarian Study"

_2073-4468, 2022, doi:10.3390/antib11020023_

Round 1
Reviewer 1 Report
After reading the content of the thesis, I am writing because some corrections are needed.
- It should be explained by what criteria the 24 food antigens selected in this thesis are chosen.
- The reaction classes for classifying responses to 24 food antigens are ambiguous. The difference in the range between classes is large. It seems better to divide moderative positive(0.36~17.50) and strongly positive(17.51<) into a range rather than notation by class.
- clas 1 -> class 1(page 4. first paragraph.)
- phospectrometer(?. page 4. 21st line) - photospectrometer?
Author Response
After reading the content of the thesis, I am writing because some corrections are needed.
- Thank you for the overall evaluation of our paper as good and for critical recommendations.
- It should be explained by what criteria the 24 food antigens selected in this thesis are chosen.
- Thank you for the valuable note. We have added this in the paper, Material and methods section. We choose this extended panel based on the previous studies for the most common food antigens investigated in IBS patients.
- The reaction classes for classifying responses to 24 food antigens are ambiguous. The difference in the range between classes is large. It seems better to divide moderative positive(0.36~17.50) and strongly positive(17.51<) into a range rather than notation by class.
- Yes, we completely agree with the referee that in some way, the differences between classes may be not enough distinguishing. However, this classification was provided by the manufacturer and was found in the other studies, thus, we choose to keep it and to compare our results to others more effectively.
- clas 1 -> class 1(page 4. first paragraph.)
- Thank you for the valuable note. We have corrected the mistake.
- phospectrometer(?. page 4. 21st line) - photospectrometer?
- Thank you for noticing this technical mistake, we have corrected the issue.
Reviewer 2 Report
Authors have shown huge potential with this piece of article. Th article shows huge future prospects. Though authors failed to discuss the importance of the results in the discussion section a substantial revision will be required
Minor comments:
References are missing in results and discussion section.
Line 155 sequence of groups are not in proper order.
Few typos are highlighted in yellow.
Fig.1 It seems that there are two clusters in patients group, IgG hi and Low. Can authors shed some light or discuss the result further Line 197
Line 252-253 can author elaborate the paragraph further.
Line 305 mention the author name used in Reference.
Authors contributions section can be expanded with respective contributions.
Author Response
Authors have shown huge potential with this piece of article. The article shows huge future prospects. Though authors failed to discuss the importance of the results in the discussion section a substantial revision will be required.
- Thank you very much for the critical notes and for evaluating our paper as good. We agree that our discussion section may have some issues regarding the explanation of our results. Initially, we did our best to include all relevant to our research papers and to compare our results. We took your note very seriously and revise the discussion additionally and accordingly, as far as possible.
Minor comments:
References are missing in results and discussion section.
- Thank you for noticing this to us. The reason for this appearance as there are not enough references is that we put a lot of our explanations and critical thinking over our results. You can see that we cited a lot of papers similar to our investigation. Furthermore, the statements of the American Academy of Allergy, Asthma and Immunology (AAAAI), the Canadian Society of Allergy and Clinical Immunology (CSACI) and the European Academy of Allergy and Clinical Immunology (EAACI), were also included and cited.
Line 155 sequence of groups are not in proper order.
- Thank you very much for this note, we corrected it.
Few typos are highlighted in yellow.
- Thank you for the help with this. Additionally, we went over the whole manuscript to ensure that it is typos-free.
Fig.1 It seems that there are two clusters in the patients group, IgG hi and Low. Can authors shed some light or discuss the result further Line 197
- Thank you for the valuable suggestions. Indeed, there is a kind of two clusters – above and below the median. However, no association or other observations were found. We added this information in the discussion section, as you recommended.
Line 252-253 can author elaborate the paragraph further.
- Thank you for the critical note, we agree completely with the referee and extended the paragraph, as recommended.
Line 305 mention the author name used in Reference.
- Thank you for the great suggestion. We have cited the name of the leading investigator, as their contribution to the field is significant.
Authors contributions section can be expanded with respective contributions.
- Thank you for the valuable note. During the submission system, we put this information, but we forgot to paste it into the manuscript. We have corrected the mentioned issue.
Reviewer 3 Report
The paper describes the serum levels of IgG, IgE and increased zonulin in patients with irritable bowel syndrome (IBS).
The paper is well organised and results are well exposed and discussed. The increased IgG levels to specific foods in this patients was previously published as the authors estate. It would be interesting to study the 4 subtypes of IgG separately in patients and controls, they have different properties and could shed light about immune disregulation that is happening in (IBS).
Specific comments:
Figure 3 is unclear. I suggest to separate IgE and IgG in two panels and specify were each p come from. Besides, egg white IgG is invisible, and zonulin is unnecessary here as it is in figure 2 and there is no association.
Author Response
The paper is well organised and results are well exposed and discussed. The increased IgG levels to specific foods in this patients was previously published as the authors estate. It would be interesting to study the 4 subtypes of IgG separately in patients and controls, they have different properties and could shed light about immune disregulation that is happening in (IBS).
- Thank you for the valuable notes and the overall evaluation of our paper a good.
Specific comments:
Figure 3 is unclear. I suggest to separate IgE and IgG in two panels and specify were each p come from. Besides, egg white IgG is invisible, and zonulin is unnecessary here as it is in figure 2 and there is no association.
- Thank you for the critical note. We acknowledge that the figure is difficult for understanding since we put data on three dimensions. We completely agree with the referee that zonulin levels are presented in figure 2, and that grouping IgE and IgE as two panels would be more representative.
- However, we would suggest to the reviewer and editor to leave the figure in such a way, because we can compare on one figure specific antibodies and zonulin levels at two dimensions. Additionally, the food antigens now are grouped as IgE (triangles) and IgG (squares), and we are afraid that if we put the data into two panels, it wouldn`t be comprehensive enough and might look like insufficient data.